# Impact of COVID-19 on the Health-Related Quality of Life of Patients during Infection and after Recovery in Saudi Arabia

**DOI:** 10.3390/ijerph20065026

**Published:** 2023-03-12

**Authors:** Menyfah Q. Alanazi, Waleed Abdelgawwad, Thamer A. Almangour, Fatma Mostafa, Mona Almuheed

**Affiliations:** 1Drug Policy & Economic Centre, King Abdulaziz Medical City, Ministry of National Guard-Health Affairs, Riyadh 11426, Saudi Arabia; 2King Abdullah International Medical Research Centre (KAIMRC), King Saud Bin Abdulaziz University for Health Sciences (KSAU-HS), Riyadh 11426, Saudi Arabia; 3College of Management, Midocean University, Moroni 6063, Comoros; 4Department of Clinical Pharmacy, College of Pharmacy, King Saud University, Riyadh 11451, Saudi Arabia; 5National Center for Artificial Intelligence (NCAI), Riyadh 12391, Saudi Arabia

**Keywords:** COVID-19, EQ-5D-5L, health-related quality of life, Saudi Arabia

## Abstract

This study evaluated the impact of COVID-19 and other factors on the health-related quality of life (HRQoL) of Saudi patients during infection and after recovery using the EQ-5D-5L and EQ-VAS instruments. An observational prospective study was conducted in November 2022, during which 389 COVID-19 patients were surveyed during their visit to a medical center. Two weeks after their recovery, they were contacted again to re-evaluate their HRQoL (192 patients either refused to participate or withdrew). The mean of the EQ-5D-5L index and EQ-VAS scores significantly increased from (0.69 ± 0.29 and 63.16 ± 24.9) during infection to (0.92 ± 0.14 and 86.96 ± 15.3) after recovery. Specifically, COVID-19 patients experienced improvement of several HRQoL dimensions post recovery, such as better mobility, enhanced self-care, returning to usual activities, less pain/discomfort, and alleviated anxiety/depression. Multiple linear regression analyses showed that having a normal weight, being employed, not being anemic, and previously taking the BCG vaccine were positively associated with a greater change in the HRQoL. An interaction between being asthmatic and taking the influenza vaccine positively predicted a lower change in the HRQoL. Having a normal weight positively predicted a greater change in the perceived health state after recovery. Increasing the consumption of natural supplements (honey and curcuma) did not improve the HRQoL or the perceived health state. Based on these findings, COVID-19 mildly impacted the HRQoL of Saudis with varying effects depending on some socio-demographic/clinical characteristics of the patients.

## 1. Introduction

Coronavirus disease 2019 (COVID-19) is an infectious acute respiratory disease caused by the novel coronavirus SARS-CoV-2 [1]. Soon after the first case was reported in 2019, this highly infectious virus rapidly became a pandemic [1,2]. The severity of COVID-19 ranges from enduring mild to severe symptoms to pneumonia, acute respiratory distress syndrome (ARDS), multi-organ dysfunctions, and death [2,3,4]. Aside from the clinical consequences, the impact of COVID-19 on the quality of life extends further [5]. In the Arabian Gulf region, the Kingdom of Saudi Arabia was witnessing a surge of a MERS-CoV epidemic when the first COVID-19 case was confirmed on 2 March 2020. As of 2 November 2022, SARS-CoV-2 has infected more than 822 thousand Saudi Arabian residents and killed more than 9400 [6].

Previous studies mainly focused on the epidemiology and transmission of SARS-CoV-2, infection control and prevention, COVID-19 vaccination, disease burden, and treatment options. However, little effort was made to evaluate the impact of COVID-19 on the health-related quality of life (HRQoL) of Saudi Arabian residents infected by SARS-CoV-2 [7,8]. Quality of life (QoL) was conceptualized as a broad multidimensional subjective evaluation of an individual’s various life aspects [9]. The World Health Organization (WHO) referred to it as an individual’s perception of their position in life, an extension of the environment in which they live, and in relation to their goals, expectations, standards, and concerns [9,10]. Quality of life (QoL) is an important goal of treatment in chronic illnesses, and is also used to identify the range of problems that can affect patients [9,10].

HRQoL is a more specific measure that quantifies the physical and mental status of an individual or a group over time. This metric is generally used to represent the impact of an illness and its management on an individual’s ability to live a fulfilling life and their overall life satisfaction [11]. HRQoL is an essential indicator of the burden of a disease that helps clinicians and policy-makers optimize patient care and public health decisions [11,12].

Many instruments have been implemented to measure HRQoL, including the five-dimensional EuroQol (EQ-5D-5L). The EQ-5D-5L is one of the most frequently used, preference-based measures of HRQoL worldwide that evaluates the HRQoL based on the patient’s perspective of their health [13,14,15,16] EQ-5D-5L is a valid and reliable measure that has been applied to countless disease areas and is the most commonly used tool in cost–utility analyses and for appraising healthcare interventions [13,16,17,18,19,20]. Since its development, it has been validated across various settings and among different populations. However, in terms of COVID-19 and in conservative communities such as that in Saudi Arabia, the use of EQ-5D-5L as a measure of HRQoL was scarce [16,19].

Being a novel virus and a large-scale pandemic, it is important to evaluate HRQoL among individuals within the context of COVID-19 not only during the disease, but also after recovery. The individual’s experience with COVID-19 symptoms of various severities, quarantine, exposure to news on COVID-19 morbidity/mortality, and social stigmatization is unique. Exploring all the possible factors that positively impact the HRQoL during COVID-19 and proposing tailored interventions will lead to a better and an expedited recovery. In Saudi Arabia, there is a paucity of research on the impact of COVID-19 on HRQoL. Moreover, it is not clear what the possible factors are that would enhance HRQoL after recovery from COVID-19.

Factors associated with HRQOL in chronic diseases such as diabetes, breast cancer, arthritis, and hypertension are commonly investigated [7,8]. For instance, improving certain lifestyle characteristics such as weight reduction, physical activity, and smoking cessation were associated with better HRQoL in such diseases. In terms of infectious diseases such as COVID-19, the experience might be different as other factors might play a role, such as previous vaccination (COVID-19, seasonal flu) and consumption of natural supplements such as drinking herbal drinks, consuming honey, or others which COVID-19 patients might perceive to be beneficial. Studies showed that herbal supplements and honey—which are commonly used in Saudi Arabia—have some antiviral activities [20,21,22,23]. In terms of vaccination, exploring the change in HRQoL during COVID-19 and after recovery can aid in persuading vaccine-hesitant groups to get vaccinated. In terms of consuming natural supplements, sharing personal experiences can inform researchers, policymakers, and other individuals at risk of contracting COVID-19.

The aim of this study was to evaluate the impact of COVID-19 and other factors on the HRQoL of Saudi patients during infection and after recovery. We hypothesize that the mean difference in the HRQoL scores (during the disease and after recovery) might be associated with certain socio-demographics, health-related factors, vaccination status, and other self-reported complementary health interventions that participants perceive to be beneficial.

## 2. Methods

### 2.1. Study Design

This was an observational prospective study, in which a cohort of confirmed COVID-19 patients who visited the influenza clinic at King Abdulaziz Medical City (KAMC), Riyadh, Saudi Arabia, were enrolled between November and December of 2022. For convenience, COVID-19-positive cases circumstantially present at the targeted setting during the study period were screened for eligibility then invited to participate. Baseline measures were obtained at the clinic and then repeated after two weeks of recovery.

### 2.2. Study Setting

This study was conducted in the influenza clinic of KAMC, a 1505-bed university-affiliated tertiary care center, accredited by the Joint Commission International.

### 2.3. Study Population

The participants were all adults between 18 and 59 years of age with a confirmed diagnosis of COVID-19, based on a rapid antigen test (RAT). Those who agreed to participate in this study provided informed consent. Participants who were severely ill and admitted to the hospital were excluded. Patients unable to read and write were also excluded from this study.

### 2.4. Data Collection

A self-administered questionnaire was initially used in this study at baseline, followed by an online survey after two weeks of recovery. At baseline, the questionnaire was coupled with a letter of invitation and an informed consent, all in Arabic. The questionnaire was handed to participants by the research investigators who assisted them when needed. The study participants were provided with a Google Survey link to be accessed and filled in two weeks after of discharge. The research investigators were familiar with the targeted setting and population in terms of language and cultural norms. The questionnaire consisted of:

#### 2.4.1. Study Exposures

Socio-demographics included sex, age (years), level of education (higher education, middle education, low education), marital status (single, married, other), and employment status (employed, unemployed, student).

Clinical or health-related characteristics included the previous medical history (having a chronic disease), previous vaccination history against seasonal influenza, Bacillus Calmette–Guérin (BCG), and COVID-19, smoking status (current smoker/never smoked/former smoker), body mass index (obesity: BMI ≥ 30 kg/m^2^), and any natural supplement (dietary) consumed during COVID-19 onset to overcome the discomfort of the disease. The symptoms and duration of COVID-19 infection were also recorded.

#### 2.4.2. Study Outcomes

After obtaining prior approval from the EuroQol Research Foundation, we utilized the EQ-5D-5L tool [16]. Several studies have confirmed the validity and reliability of the Arabic EQ-5D-5L in different Arabian populations, and the tool has proved to be a valid measure for HRQoL in Arabic speaking populations. The EQ-5D-5L instrument has two components: a description of the health state (EQ-5D-5L descriptive system) and a self-evaluation or perception of the individual’s health state using a visual analog scale (EQ-VAS). The EQ-5D-5L consists of five dimensions: mobility, self-care, usual activities, pain/discomfort, and anxiety/depression. Each dimension has five levels of severity (no problems: 1, slight problems: 2, moderate problems: 3, severe problems: 4, extreme problems: 5. The responses are combined to produce a five-digit number describing the respondent’s health status (ranging from 11,111 to 55,555): 11,111 (having no problems in all dimensions, or full health) and 55,555 (having extreme problems in all dimensions, or worst health). The EQ-5D-5L scores are then standardized to become an index ranging from 0 (representing death) to 1 (representing full health), with negative values representing states worse than death [13,16,17]. The second part of the tool is a visual vertical scale ranging from “0” to “100” ranging from “Best imaginable health state” (100) to “Worst imaginable health state” (0) [13,16,17,18,19,20].

### 2.5. Data Management and Analysis

Statistical analyses were performed using SPSS (version 25; SPSS Inc., New York, NY, USA). Descriptive statistics of categorical variables were presented in frequency and percentage, while continuous variables were presented in mean ± SD, or median (interquartile range). Measures of internal consistency showed that the Cronbach’s alpha of EQ-5D-5L during COVID-19 and after recovery were 0.842 and 0.801, respectively.

Same-group analyses (within subgroups) using the Wilcoxon signed ranks test was performed to compare the index scores of EQ-5D-5L dimensions and EQ-VAS during the disease and after recovery; given the distribution of EQ-5D-5L, data were ordinal and commonly skewed. The assumptions of analyses of covariance (ANCOVA) were violated due to the nature of skewed outcomes and the presence of logic outliers. Therefore, between-group analyses were performed using the Mann–Whitney test by comparing the differences in the EQ-5D-5L scores and EQ-VAS (recovery scores minus baseline scores). Higher differences in scores indicated a greater change in the HRQoL.

Variables statistically significant at the bivariate level of analyses (*p* < 0.05) were counted as potential predictors of change in EQ-5D-5L and EQ-VAS scores. Accordingly, two multiple linear regression models were constructed. To meet the regression assumptions, the skewness in the EQ-5D-5L scores was corrected by normalization (scores minus the minimum value/range) to take a range from zero to one and then log10 transformed (skewness coefficient became −0.150). Homoscedasticity and normality of the data were confirmed after examining the P–P plot and scatterplot. An absence of multi-collinearity was confirmed by examining the variance inflation factor (VIF) values, which were all under 5, indicating that the assumption is met. In terms of EQ-5D-5L scores, no statistically significant interactions between various exposures were detected, except for being asthmatic and taking the seasonal influenza vaccine. The model was fit, and the adjusted R^2^ was 33.5%. Insignificant variables such as curcuma/onion/peppermint/wild thyme consumption, zinc deficiency, sleep duration, diabetes, and gender were dropped out, which increased the adjusted R^2^ to 34.2%. In terms of EQ-VAS scores, no significant interaction effect was present. The model was fit, and the adjusted R^2^ was 33.5%. Insignificant variables such as onion/peppermint/wild thyme consumption, smoking, sleep duration, asthma, anemia, gender, age, and BCG vaccination were dropped out, which increased the adjusted R^2^ to 38%. For all statistical tests, *p* < 0.05 was considered statistically significant.

### 2.6. Ethical Issues

This study was approved by the Research Committee of King Abdullah International Medical Research Center (KAIMRC), King Saud Bin-Abdulaziz University for Health Sciences (IRB/2666/22). Patient confidentiality and privacy were secured by the principal investigator.

### 2.7. Sample Characteristics

A total of 389 out of 590 Saudi patients were enrolled in this study and completed both questionnaires (response rate was 65.9%). Two thirds of the sample were women (*n* = 259). They had a mean of age 34.87 ± 8.36 years. The majority of the participants had a high level of educational (82.5%). More than two thirds of the participants were employed (67.4%), and married (64.3%). The consumption of natural supplements during COVID-19 infection was reported by 55% of the participants (curcuma 28.5%, peppermint 25.7%, and others). Increased consumption of honey was reported by 53% of the participants, followed by an increased consumption of onion (34.7%) and garlic (33.2%) (Table 1).

When asked about the number of times they had an episode of cold symptoms in the past six months, 352 (90.5%) participants reported having one to three episodes, yet none were confirmed to be COVID-19 by a RAT test. Almost half of the participants (55.8%) had at least one chronic disease. The most common disease reported by study participants was vitamin-D deficiency (30.8%). Thirty percent of the patients were classified as obese. The self-reported symptoms of COVID-19 included fatigue (66.1%), fever (48%), cough (35.5%), anxiety/depression (34.4%), pain (46%), numbness (5.4%), and itching (4.6%). The most common persisting symptoms after recovery were cough (14.9%), diminished sense of taste (13.6%), diminished sense of smell (12.3%), fatigue (9.8%), and auditory dysfunction (2.4%) (Table 2).

## 3. Results

### 3.1. Assessment of HRQoL and Perceived Health Status

The EQ-5D-5L index scores significantly increased from 0.68 ± 0.31 during COVID-19 to 0.92 ± 0.13 after 2 weeks of recovery (*p* < 0.001 *). The severity of problems related to mobility, self-care, usual activities, pain/discomfort, and anxiety/depression (the five dimensions of EQ-5D-5L) all significantly decreased after recovery, which indicated that the overall quality of life of COVID-19 patients was enhanced due to recovery (*p* < 0.001 each). Similar improvements were observed in terms of EQ-VAS scores, as patients’ score increased from 63.16 ± 24.92 during infection to 86.96 ± 15.31 after recovery (*p* < 0.001) (Table 3). However, exploration was needed to determine which factors contributed to a significant enhancement in HRQoL and perceived health status.

### 3.2. Predictors of HRQoL and Health Status

As shown in Table 4, within-group analyses showed that EQ-5D-5L index and EQ-VAS scores showed statistically significant improvement within all subgroups. However, the differences in scores (between recovery and COVID-19 infection) provided further insight into which subgroups are expected to show higher degrees of improvement compared to other groups. Between-group analyses showed that obesity, employment status, sleep duration, consumption of natural supplements (curcuma, peppermint, honey, and onion), previous medical history (asthma, anemia, zinc deficiency, and diabetes mellitus), and vaccination (influenza and BCG) were associated with changes in EQ-5D-5L index scores. For instance, the changes in EQ-5D-5L index scores were significantly greater among non-obese participants (0.37 ± 0.20), employed participants (0.27 ± 0.28), those who usually slept < 6 h (0.30 ± 0.29), and others shown in Table 4, indicating higher degrees of improvement. In terms of changes in EQ-VAS scores, males (28.11 ± 24.22), those with normal weight (45.74 ± 0.18), employed participants (27.71 ± 25.35), and others reported an improvement in their perceived health status (Table 4).

Multiple linear regression analyses were performed to model the relationship between obesity, employment status, sleep duration, consumption of natural supplements, previous medical history, vaccination (influenza and BCG), and others with the changes in the ED-5D-5L index and EQ-VAS scores. In terms of the ED-5D-5L index, after adjusting for the effect of all variables, having a normal weight, being employed, not being anemic, and previously taking the BCG vaccine significantly contributed to a greater change in ED-5D-5L index scores. An interaction between being asthmatic and taking the influenza vaccine predicted lower changes in ED-5D-5L index scores. In terms of EQ-VAS scores, having a normal weight predicted a greater change in the EQ-VAS scores after recovery. Increasing the consumption of natural supplements (honey and curcuma) did not improve either of the study outcomes (Table 5).

## 4. Discussion

COVID-19 infection is indisputably an unpleasant experience that impacted the HR-QoL of people in Saudi Arabia. Despite being an acute respiratory disease, some Saudi Arabians might not have fully recovered in many aspects such as mobility, self-care, daily activities, body pain, and anxiety/depression. In this study, we hypothesize that the recovery from COVID-19 will naturally enhance the HR-QoL among patients, yet some factors might have contributed to a better improvement of the HR-QoL. This study is one of the few that examined these factors using the EuroQol 5-dimension, 5-level (EQ-5D-5L) questionnaire and a visual analog scale (EQ-VAS) during and post infection.

Recovery from COVID-19 enhanced the quality of life among Saudi patients, as the EQ-5D-5L index scores and EQ-VAS scores significantly increased after two weeks. This can be attributed to the absence of COVID-19 signs and symptoms. A previously published meta-analysis study stated that 58% of COVID-19 patients reported having a poor quality of life after recovery [8]. Our observed EQ-5D-5L scores after recovery were similar to figures reported in China (0.949), yet higher than those reported in Morocco (0.86), the UK (0.714), Norway (0.690), and Belgium (0.620) [24,25,26,27,28]. Saudis were able to resume their normal physical activity and fully attend to their self-care needs. Pain and discomfort due to COVID-19 signs and symptoms resolved after the COVID-19 infection, despite the presence of persistent signs/symptoms in up to 27% of our sample. Saudis were less anxious and less depressed after recovery, being able to resume their daily activities after quarantine [7,8,29,30]. Even in regard to EQ-VAS scores, Saudis reported higher scores compared to COVID-19-infected patients in Germany and Belgium [28,31]. This indicated that some variables might have played a role in the enhancement of HR-QoL besides the recovery from the disease.

Sample characteristics were tested to identify which factors led to a significant improvement in the HRQoL. For instance, being obese was associated with a lower HR-QoL change in comparison to those with normal weight. According to the World Health Organization (WHO), the overall prevalence of obesity in Saudi Arabia is estimated to be 33.7% [32,33]. The severity of COVID-19 might have been higher among obese participants, which led to a delayed recovery. Previous studies showed that COVID-19 patients who are obese tend to have less improvement in their quality of life. This finding aligns with the conclusion of a large-scale systematic review that obesity is significantly associated with increased severity and higher mortality among COVID-19 patients. Obesity is also commonly associated with hypertension, dyslipidemia, cardiac problems, diabetes, and others, all of which impair the HRQoL in its various aspects [34].

Additionally, having a chronic disease such as asthma contributed to a delayed recovery or less improvement in the HRQoL. One study showed that among 562 asthma patients, 21% were hospitalized, 3% received mechanical ventilation, and one died. Anemia was also an independent risk factor associated with the severity of COVID-19 [35]. This entails that these patients in particular are expected to have a compromised quality of life after recovery and, thus, need more support. Previous vaccination status, especially BCG and seasonal influenza vaccination, was associated with the changes in the HRQoL after recovery. COVID-19 Saudi patients who were previously vaccinated against BCG and seasonal influenza were able to report higher changes in their HRQoL. It was reported that some BCG vaccine strains can be used as an additional defense in future pandemics [36]. The WHO and more than 20 European countries have already recommended the co-administration of influenza and COVID-19 vaccines [37]. Our findings can be useful to vaccine campaigns promoting the benefits of these vaccines, especially when targeting vaccine-hesitant groups.

While questioning the Saudi study participants about consuming natural supplements during the onset of symptoms, they commonly reported an increased consumption of natural supplements such as honey, garlic/onion, and herbal drinks. The benefits of these natural supplements stem from deeply rooted cultural norms, despite the fact that their impact on HRQoL during COVID-19 infection was not rigorously tested in the past. A previous clinical trial enlisted some of honey’s antiviral and antibacterial properties, since its antioxidant content hugely impacts the co-morbidities associated with SARS-CoV-2 infection [23]. However, our study findings cannot be conclusive, as experimental study designs are required.

## 5. Limitations

This study had some limitations. First, it was conducted in only one setting, which limits its generalizability to other hospitals. Two thirds of the participants were female, which might have produced some bias. Older participants had a lower response than the younger age groups. The measurement of health status using the EQ-5D-5L instrument may have resulted in over- or underestimation of QoL because some patients might have inflated their self-assessment of their quality of health. Finally, one of the challenges in this study was the lack of a baseline measure before the exposure to COVID-19 infection. Unfortunately, recruiting a healthy group, obtaining a baseline measure of HRQoL, then awaiting COVID-19 infection is a lengthy process and prone to a loss in follow-up. Questioning COVID-19 patients about their HRQoL prior to COVID-19 infection is also prone to recall bias. Despite these limitations, factors significantly associated with changes in the HRQoL and perceived health status in this study remain valid within the context of COVID-19 infections and the Saudi Arabian population.

## 6. Conclusions

COVID-19 had a significant negative impact on Saudis’ HRQoL with varying degrees. The changes in the QoL two weeks after contracting COVID-19 were greater among individuals with normal weight, who were employed, non-anemic, and had previously taken the BCG vaccine. An interaction between being asthmatic and taking the influenza vaccine also contributed to greater changes. Further studies are needed to test the effectiveness of increased natural supplements on the changes in HR-QoL. Investigating the health status of Saudi patients with COVID-19 and identifying significantly associated factors with HRQoL will optimize future clinical approaches and inspire public health policies toward maximizing their effectiveness and efficiency. The implementation of such strategies should greatly improve COVID-19 patient outcomes and, of course, their quality of life.

## Figures and Tables

**Table 1 ijerph-20-05026-t001:** Socio-demographic and lifestyle characteristics of the sample.

Study Exposures	N (%)
Sex	
Male	130 (33.4)
Females	259 (66.6)
Age (years)	
Mean ± SD	34.9 ± 8.4
Education, years	
Low education (≤9)	12 (3.1)
Middle education (12–15)	56 (14.4)
Higher education (≥16)	321 (82.5)
Employment status	
Student	40 (10.3)
Employed	262 (67.4)
Unemployed	87 (22.4)
Marital status	
Single	118 (30.3)
Married	250 (64.3)
Other	21 (5.4)
Bodyweight	
Normal	272 (69.9)
Obese	117 (30.1)
Cigarette smoking	
Current smoker	66 (17.0)
Never smoked	259 (66.6)
Former smoker	64 (16.4)
Daily sleep hours	
Less than 5 h	122 (31.3)
From 6 to 7 h	216 (55.5)
More than 7 h	51 (13.2)
Consumption of natural supplements *	
No	175 (45)
Yes	214 (55)
Curcuma	111 (28.5)
Wild thyme	18 (4.6)
Nigella sativa	64 (16.5)
Costus	30 (7.7)
Peppermint	100 (25.7)
Apiaceae	29 (7.5)
Chamomile	20 (5.1)
Ginger	31 (8)
Change in dietary pattern	
Increased honey consumption	206 (53)
Increased garlic consumption	129 (33.2)
Increased onion consumption	135 (34.7)

*: non-mutually exclusive.

**Table 2 ijerph-20-05026-t002:** Clinical and disease-related characteristics of the sample.

Study Exposures	N (%)
Chronic health conditions	
No	172 (44.3)
Yes	217 (55.7)
Vitamin D deficiency	120 (30.8)
Asthma	62 (15.9)
Iron deficiency (anemia)	54 (13.9)
Zinc deficiency	19 (4.9)
Diabetes mellitus	21 (5.4)
Cardiovascular/hypertension	43 (11.5)
Symptoms during infection *	
Fatigue	257 (66.1)
Headache	202 (51.9)
Fever	187 (48)
Pain	179 (46)
Anorexia	144 (37)
Cough	138 (35.5)
Anxiety/depression	134 (34.4)
Sore throat	105 (27.0)
Lost sense of smell	70 (18.3)
Dyspnea	69 (17.7)
Arthralgia	66 (17.0)
Lost sense of taste	63 (16.2)
Sleeping disturbances	48 (12.3)
Hearing problem	40 (10.3)
Numbness	21 (5.4)
Itching	18 (4.6)
Vomiting/diarrhea	10 (2.7)
Sinusitis	2 (0.5)
Persistent symptoms *	
Hoarseness of voice	105 (27.0)
Anxiety/depression	70 (17.7)
Cough	58 (14.9)
Lost sense of taste	53 (13.6)
Lost sense of smell	48 (12.3)
Sleeping disturbances	44 (11.15)
Fatigue	39 (9.8)
Shortness of breath	28 (7.2)
Headache	23 (5.9)
Arthralgia	25 (6.4)
Sore throat	23 (6.0)
Dizziness	10 (2.6)
Hearing problem	9 (2.4)
Anxiety/depression	8 (2.1)
Chest pain	5 (1.3)
Pain in bones	3 (0.8)
Loss of appetite	7 (1.8)
Sinusitis	2 (0.5)
Itching	2 (0.5)
Duration of the COVID-19 infection	
Mean ± SD (days)	8.6 ± 5.0
Onset of recovery	
Mean ± SD (days)	6.22 ± 3.48
Previous vaccination	
Seasonal influenza vaccine	147 (37.8)
Bacillus Calmette–Guérin vaccine (BCG)	288 (74.0)
COVID-19 vaccine	344 (88.4)

*: non-mutually exclusive.

**Table 3 ijerph-20-05026-t003:** Perceived HRQoL and health status during COVID-19 and after recovery.

Study Outcomes	During COVID-19 Mean ± SD M[IQR]	After Recovery Mean ± SD M[IQR]	Statistical Test, *p*-Value
EQ-5D-5L index score	0.68 ± 0.31 0.8 (0.37)	0.92 ± 0.13 1 (0.09)	Z = −14.849, *p* < 0.001 *
Mobility score	2.34 ± 1.18 2 (2)	1.24 ± 0.57 1 (0)	Z = −13.36, *p* < 0.001 *
Self-care score	1.67 ± 1.08 1 (1)	1.11 ± 0.37 1 (0.37)	Z = −9.223, *p* < 0.001 *
Usual activity score	2.24 ± 1.35 2 (2)	1.35 ± 0.72 1 (0)	Z = −11.348, *p* < 0.001 *
Pain/discomfort score	2.47 ± 1.27 2 (2)	1.45 ± 0.78 1 (1)	Z = −12.041, *p* < 0.001 *
Anxiety/depression score	2.25 ± 1.31 2 (2)	1.6 ± 0.94 1 (1)	Z = −9.775, *p* < 0.001 *
EQ-VAS score	63.2 ± 24.9 70 (40)	86.96 ± 15.31 90 (20)	Z = −13.87, *p* < 0.001 *

Abbreviations: EQ-5D-5L index score: 0 (representing death) to 1 (representing full health); EQ-5D-5L dimension scores range from level I to V: mild to extreme problems; EQ-VAS score range from 0 to 100: best imaginable health state to worst; SD: standard deviation; M: median; Z: Wilcoxon test, *: *p* value statistically significant at <0.05.

**Table 4 ijerph-20-05026-t004:** EQ-5D-5L and EQ-VAS scores within and between subgroups.

Exposures	EQ-5D-5L Index Scores			EQ-VAS Scores		
	During Infection	After Recovery	Difference in Scores	During Infection	After Recovery	Difference in Scores
	Mean ± SD	Mean ± SD	Mean ± SD	Mean ± SD	Mean ± SD	Mean ± SD
Sex						
Males	0.71 ± 0.27	0.96 ± 0.08 *	0.25 ± 0.26	62.8 ± 23.6	90.9 ± 11.6 *	28.11 ± 24.22 **
Females	0.68 ± 0.31	0.9 ± 0.15 *	0.23 ± 0.27	63.3 ±25.6	84.9 ± 16.5 *	21.6 ± 25.13
Age (years)						
19–39	0.68 ± 0.29	0.93 ± 1.34 *	0.24 ± 0.25	63.73 ± 25.6	87.86 ± 15.79 *	24.12 ± 25.74
40–59	0.67 ± 0.34	0.92 ± 0.12 *	0.25 ± 0.31	61.71 ± 23.1	84.68 ± 13.84 *	22.97 ± 23.05
Obesity						
Normal	0.72 ± 0.32	0.97 ± 0.07 *	0.37 ± 0.20 **	68.9 ± 25.2	83 ± 15.9 *	45.74 ± 0.18 **
Obese	0.59 ± 0.22	0.90 ± 0.15 *	0.18 ± 0.28	49.8 ± 18.2	95.5 ± 8.8 *	14.13 ± 21.06
Education						
Low/middle education	0.68 ± 0.27	0.89 ± 0.16 *	0.22 ± 0.22	63.82 ± 21.19	82.56 ± 17.41*	18.73 ± 24.76
Higher education	0.69 ± 0.3	0.92 ± 0.13 *	0.25 ± 0.28	63 ± 25.6	87.9 ± 14.6 *	24.88 ± 24.94
Employment status						
Employed	0.67 ± 0.29	0.93 ± 0.12 *	0.27 ± 0.28 **	60.3 ± 24.9	88 ± 15.3 *	27.71 ± 25.35 **
Unemployed/student	0.72 ± 0.31	0.90 ± 0.16 *	0.18 ± 0.24	68.94 ± 23.97	84.69 ± 15.06 *	15.74 ± 20.68
Marital status						
Single/divorced/widow	0.67 ± 0.33	0.91 ± 0.14 *	0.25 ± 0.27	59.21 ± 26.09	85.59 ± 16.00 *	26.38 ± 27.11
Married	0.69 ± 0.28	0.92 ± 0.13 *	0.24 ± 0.27	65.3 ± 24	87.7 ± 14.8 *	22.37 ± 23.66
Daily sleep hours						
<6 h	0.62 ± 0.34	0.92 ± 0.14 *	0.30 ± 0.29 **	58.39 ± 26.18	87.16 ± 14.9 *	28.76 ± 25.64 **
≥6 h	0.72 ± 0.29	0.93 ± 0.13 *	0.21 ± 0.26	65.34 ± 24.06	86.88 ± 15.52 *	21.54 ± 24.40
Smoking						
Yes	0.74 ± 0.25	0.96 ± 0.07 *	0.22 ± 0.25	60.6 ± 24.85	91.09 ± 11.32 *	30.53 ± 23.89 **
No	0.67 ± 0.32	0.92 ± 0.14 *	0.25 ± 0.28	63.69 ± 24.94	86.12 ± 15.89 *	22.42 ± 25.01
Natural supplements						
Curcuma	0.74 ± 0.27	0.93± 0.13 *	0.18 ± 0.23 **	72.9 ± 21.9	86.4 ± 15.9 *	13.52 ± 19.44 **
Wild thyme	0.73 ± 0.24	0.91 ± 0.11 *	0.25 ± 0.27	63.2 ± 26.1	89.6 ±13.5 *	23.68 ± 24.78
Costus	0.63 ± 0.3	0.87 ± 0.17 *	0.24 ± 0.26	68.9 ± 27.7	86.6 ± 15.7 *	17.70 ± 26.34
Peppermint	0.73 ± 0.27	0.91 ± 0.14 *	0.17 ± 0.21 **	70.6 ± 22.2	84.4 ± 15 *	13.80 ± 19.21 **
Honey	0.83 ± 0.15	0.94 ± 0.1 *	0.11 ± 0.16 **	72.1 ± 22.4	86.5 ± 14.4 *	14.40 ± 21.76 **
Garlic	0.75 ± 0.28	0.91 ± 0.14 *	0.27 ± 0.29	70.9 ± 23.5	85 ± 14.7 *	26.42 ± 26.97
Onion	0.73 ± 0.28	0.92 ± 0.13 *	0.19 ± 0.23 **	67.9 ± 25.1	85.8 ± 14.8 *	17.91 ± 22.11 **
Medical history						
Vitamin D deficiency	0.62 ± 0.33	0.89 ± 0.17 *	0.27 ± 0.29	58.6 ± 27.4	83.9 ± 18. 5*	25.26 ± 24.84
Asthma	0.56 ± 0.36	0.87 ± 0.2 *	0.33 ± 0.29 **	56.4 ± 28.1	85.1 ± 19.3 *	28.68 ± 27.64
Anemia	0.51	0.86 ± 0.19 *	0.36 ± 0.31 **	54.2 ± 27.2	83.6 ± 15.8 *	29.52 ± 20.61 **
Zinc deficiency	0.59 ± 0.27	0.95 ± 0.1 *	0.37 ± 0.23 **	53.1 ± 23.5	96.3 ± 10.1 *	43.16 ± 20.83 **
DM	0.5 ± 0.39	0.87 ± 0.17 *	0.39 ± 0.38 **	53.2 ± 23.7	83.1 ± 15.2 *	29.90 ± 24.78
Cardiovascular	0.54 ± 0.34	0.87 ± 0.13 *	0.34 ± 0.35	51.6 ± 14.5	82 ± 14.7 *	30.4 ± 24.49
Vaccinations						
Influenza	0.72 ± 0.3	0.92 ± 0.14 *	0.21 ± 0.29 **	69.5 ± 24	88.1 ± 12.9 *	18.61 ± 21.87 **
BCG	0.65 ± 0.3	0.91 ± 0.14 *	0.27 ± 0.29 **	61.4 ± 24.9	87.5 ± 15.5 *	26.14 ± 25.30 **
COVID-19	0.7 ± 0.27	0.93 ± 0.12 *	0.24 ± 0.26	64.1 ± 24.2	88.1 ± 14.7 *	24.01 ± 24.67

DM: diabetes mellitus; *: statistically significant at *p* < 0.01 using the Wilcoxon test; **: statistically significant at *p* < 0.05 using the Mann–Whitney test.

**Table 5 ijerph-20-05026-t005:** Predictors of changes in EQ-5D-5L Index and EQ-VAS scores.

Change in EQ-5D-5L Index Scores	B	Std. Error	*t*	Adj. *p*-Value
Obesity Normal vs. obese	0.094	0.025	3.757	<0.001 *
Employment status Employed vs. unemployed/student	0.059	0.023	2.575	0.010 *
Smoking Non-smoker vs. smoker	0.050	0.027	1.829	0.068
Complementary health interventions No honey vs. honey	0.182	0.022	8.403	<0.001 *
Medical history				
No anemia vs. anemia	−0.057	0.029	−1.977	0.049 *
No asthma vs. asthma	0.207	0.091	2.284	0.023 *
Vaccination status				
No BCG vaccination vs. BCG vaccination	−0.064	0.023	2.284	0.023 *
No flu vaccination vs. flu vaccination	0.295	0.102	2.911	0.004 *
Interactions Asthma and flu vaccination	−0.141	0.054	−2.612	0.009 *
Change in EQ-VAS scores				
Obesity Normal vs. obese	25.614	2.442	10.490	<0.001 *
Complimentary health interventions	B	Std. Error	*t*	Adj. *p*-value
No curcuma vs. curcuma	5.367	2.305	2.328	0.020 *
No honey vs. honey	9.192	2.195	4.188	<0.001 *
Vaccination status No flu vaccination vs. flu vaccination	3.627	2.085	1.740	0.083

B: coefficient of determination; Std. Error: standard error; *t*: Student’s *t*-test; Adj: adjusted; *: *p* value statistically significant at <0.05.

## Data Availability

Not applicable.

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
