# Peer review of "Impact of COVID-19 on the Health-Related Quality of Life of Patients during Infection and after Recovery in Saudi Arabia"

_ijerph, 2023, doi:10.3390/ijerph20065026_

Round 1

Reviewer 1 Report

Thank you for the opportunity to review this work which presents a study on the quality of life on COVID-19 patients during and post infection in Saudi Arabia. The strengths of the present work regard the measurement of health-related quality of life during and after virus infection and the examination of a series of socio-demographic and clinical factors (including the use of alternative homeopathic medicine) on quality of life dimensions in a sample of Saudi Arabian patients. The methods are clearly described however a number of issues need to be addressed before the paper is considered for publication:

Title

 The title is a bit confusing as it refers to non-hospitalized patients, this needs to be clarified. (see comment below).

 Abstract

 I suggest authors narratively present results and refrain from reporting statistics in the abstract.

Introduction

 Some passages could be eliminated as redundant and unrelated to the scope of the present paper. For instance “Saudi Arabia's economy decreased by 7% in the second quarter after the initial SARS-CoV-2 outbreak. Both the non-government and government sectors have recorded negative growth rates (10.1% and 3.5%, respectively) [5]."

Method

-The study design is not cross-sectional as I believe data were collected at two time points, during and after virus infection. This needs to be rectified also in the limitations section.

 -It is not clear what king of population group the study includes as authors seem to used different terms throughout the text to refer to their participants as patients, out-patients, non-hospitalized patients. This needs to be clarified. Also, it is not clear how and where were they recruited and what was the procedure.

Statistical Analysis.

It is not clear what kind of analysis, apart from the descriptive ones were performed on the data. This section needs to be elaborated into more detail.

Results

Authors should avoid duplication of reporting in text and in tables. For instance, all the statistical results reported in section 3.2. Socio-Demographic/Clinical Characteristics are already reported in Tables 1 and 2.

 Tables and Figures need to be improved. Tables 2, 3, 4, 5, are impossible to read probably due to formatting problems.

Figures 1 and 2 lack essential information (for instance what does the percentage refers to?).  Information on statistically significant comparisons should be introduced. Figure 2 only presents 3 out of 5 EQ-5D. Other figures could also be improved

Author Response

Response to Reviewers

(Journal: International Journal of Environmental Research and Public Health)

Study title: Impact of COVID-19 on the Health-Related Quality of Life of patients during infection and after recovery in Saudi Arabia

Dear valued reviewers,

On behalf of my fellow co-authors, we would like to acknowledge your efforts and time spent in revising our submitted paper. We strongly believe that such comments will strongly contribute to the scientific quality of the study. Please note that the changes based on your comments are highlighted in yellow.

Response to the suggested comments:

Reviwer1

Thank you for the opportunity to review this work which presents a study on the quality of life on COVID-19 patients during and post infection in Saudi Arabia. The strengths of the present work regard the measurement of health-related quality of life during and after virus infection and the examination of a series of socio-demographic and clinical factors (including the use of alternative homeopathic medicine) on quality of life dimensions in a sample of Saudi Arabian patients. The methods are clearly described however a number of issues need to be addressed before the paper is considered for publication:

Title

 The title is a bit confusing as it refers to non-hospitalized patients, this needs to be clarified. (see comment below).

Response: Apologies for this confusion, we have edited and shortened the title.

"Impact of COVID-19 on the Health-Related Quality of Life of patients during infection and after recovery in Saudi Arabia"

 Abstract

 I suggest authors narratively present results and refrain from reporting statistics in the abstract.

Response: We agree on this point. We have revised the results of the abstract as advised:

"The mean of EQ-5D-5L index and EQ-VAS scores significantly increased from (0.69 ± 0.29 and 63.16 ± 24.9) during infection to (0.92 ± 0.14 and 86.96 ± 15.3) after recovery. In specific, COVID-19 patients enhanced on several HRQoL dimensions post recovery, such as better mobility, enhanced self-care, resorting to usual activities, less pain/discomfort, and alleviated anxiety/depression".

Introduction

 Some passages could be eliminated as redundant and unrelated to the scope of the present paper. For instance “Saudi Arabia's economy decreased by 7% in the second quarter after the initial SARS-CoV-2 outbreak. Both the non-government and government sectors have recorded negative growth rates (10.1% and 3.5%, respectively) [5]."

Response: We have dropped this sentence, as well as the part  speaking on the global rates of COVID-19. We have revised the whole introduction section to make it clear.

Method

-The study design is not cross-sectional as I believe data were collected at two time points, during and after virus infection. This needs to be rectified also in the limitations section.

Response: Yes we agree. The study design is actually an observational prospective study during which data was collected at baseline during the clinic visit and at recovery (after 2 weeks).

-It is not clear what king of population group the study includes as authors seem to used different terms throughout the text to refer to their participants as patients, out-patients, non-hospitalized patients. This needs to be clarified. Also, it is not clear how and where were they recruited and what was the procedure.

Response: Please note that we revised the methods section. This was an observational prospective study, in which a cohort of COVID-19 confirmed patients who visited the influenza clinic at King Abdulaziz Medical City (KAMC), Riyadh, Saudi Arabia, were enrolled by convenience between November and December of 2022. The term "patients" was used revised to make it consistent across the study.  Baseline measures were obtained at the clinic and then repeated after two weeks of recovery. This study was conducted in the influenza clinic of KAMC, a 1505-bed university-affiliated tertiary care center, accredited by the Joint Commission International. The participants were all adults between 18 and 59 years of age with a confirmed diagnosis of COVID-19, based on a Rapid antigen test (RAT). Those who agreed to participate in this study provided an informed consent. Patients who were severely ill and admitted to the hospital were excluded. Patients unable to read and write were also excluded from this study. A self-administered questionnaire was initially used in this study at baseline, followed by an online survey after two weeks of recovery.

Statistical Analysis.

It is not clear what kind of analysis, apart from the descriptive ones were performed on the data. This section needs to be elaborated into more detail.

Results

Authors should avoid duplication of reporting in text and in tables. For instance, all the statistical results reported in section 3.2. Socio-Demographic/Clinical Characteristics are already reported in Tables 1 and 2.

 Tables and Figures need to be improved. Tables 2, 3, 4, 5, are impossible to read probably due to formatting problems

Figures 1 and 2 lack essential information (for instance what does the percentage refers to?).  Information on statistically significant comparisons should be introduced. Figure 2 only presents 3 out of 5 EQ-5D. Other figures could also be improved

Response: Please note that our data was revisited and we sought statistical consultation. Descriptive statistics of categorical variables were presented in frequency and percentage, while continuous variables were presented in mean ± SD or median (inter-quartile range). Measures of internal consistency showed that the Cronbach's alpha of EQ-5D tool during COVID-19 and after recovery were 0.842 and 0.801 respectively. Same group analyses (within subgroups) using the Wilcoxon Matched-Pairs Signed Ranks Test was performed to compare the index scores of EQ-5D dimensions and EQ-VAS during the disease and at recovery given that the distribution of EQ-5D data is ordinal and commonly skewed (Arab-Zozani, 2020). Skewness coefficients ranged between 0.5 and 3.4. The assumptions of Analyses of Covariance (ANCOVA) were violated due to the nature of skewed outcome and the presence of logic outliers. Therefore, between group analyses were performed using the Mann Whitney test by comparing the differences in the EQ-5D scores and EQ-VAS (recovery minus baseline). Higher differences in scores indicated a higher change in the HRQoL. Variables statistically significant at the bivariate level of analyses (P<0.05) were accounted as potential predictors of changes in EQ-5D and EQ-VAS scores. Accordingly, two multiple linear regression models were constructed. To meet the regression assumptions, the skewness in EQ-5D scores was corrected by normalization (score-minimum value/range) to take a range zero-one and then log10 transformed (skewness coefficient became -0.150).  Homoscedasticity and normality of the data were confirmed after examining the P-P plot and scatter plot. An absence of multi-collinearity was confirmed by examining the VIF values which were all under 5, indicating that the assumption is met. In terms of EQ-5D scores, no statistically significant interaction effects between various exposures were detected, except for being asthmatic and taking seasonal influenza vaccine. The model was fit, and the adjusted R2 was 33.5%. Insignificant variables such as curcuma/onion/peppermint/ wild thyme consumption, zinc deficiency, sleeping duration, diabetes, gender were dropped out which increased the adjusted R2 to 34.2%. In terms of EQ-VAS scores, no significant interaction effect was present. The model was fit, and the adjusted R2 was 33.5%. Insignificant variables such as onion/peppermint/ wild thyme consumption, smoking, sleeping duration, asthma, anaemia, gender, age, and BCG vaccination were dropped out which increased the adjusted R2 to 38%. Table count was dropped to five and all figures were removed.

Reviewer 2 Report

This study offers a modest added value to our understanding of patient experience and outcomes when coping with Covid-19. I prresent the following comments to help the authors bring the manuscritp to a point where it is ready for publication:

1. Language editing - while generally clearly phrased the manuscript can ebenfit from having a professional english language editor "polishing" minor grammatical errors and cumbersome phrasing. 

2. I would reconsider the claim that COVID19 has become the most severe threat of our time. While socio-economic threats due to lockdowns were quite severe the pandemic was hardly a mass killer, with mortality rates lower than heart disease and cancer. 

3. In the introduction I would like to see a more thorough review of the concept of QOL and its importnan ce which are touched on very briefly in this current version. Then I'd liek to see a description of the EQ-5D, its rationale and structure, information about reliability and validity. 

4. In the last paragraph of the intro the authors touch numerous issues (natural sup0plements, nurtrition, other vaccines etc.) - but all of these factors REQUIRE a rationale - why are they important to include in your study model? Please add these factors as variables in your literature review and refer to why would they be interesting factors to include in your research model. 

5. The main methodological challenge here is that the authors try to assess the effect of CVID-19 infection on QOL without having a baseline measure. They use a 'post infection' measure as a substitute, but this is of course still a problem. At the very least the authors must discuss this issue and detail a rationale as to why using this alternative approach will be sufficient. 

I would expect of course that most patioants with the diagnosis will report medium to low QOL - symptoms, hospitalization and the "bad reputation" of the disease alone are good reasons for that. I would absolutely expect a rise in QOL measure after gettijng better - but was QOL better before the disease? I really cannot tell. The increase in QOL may be an effect of the recovery. 

6. Re-order the results: start with descriptive statistics for the sample: mease, sds, ranges, and reliability coefficients should be presented

Then - present the comparisons between during-post disease: simple differences and while controlling for background information (gender, age, education level,  medical hitory, for example)

Then test your additional questions: the use of natural supplements, etc. 

The current propostion tables are useless. Use ANOVA and ANCOVAto test for differences. 

Use multiple regression analysis to test for the eeffects of age, background diagnoses, gender and education on QOL during and after the disease. 

Drop the excessive tables and the graphs. Present the above mentioned analyses instead. 

7. Your discussion of your results is not fully based on yourt analyses. You used descriptive analyses and then draw conclusions as to associations and effects. This is unacceptable in scientific investigation. Use hypotheses testing as mentioned above to see if the effects you were lookign for are indeed present and are statistically significant. 

Author Response

Response to Reviewers

(Journal: International Journal of Environmental Research and Public Health)

Study title: Impact of COVID-19 on the Health-Related Quality of Life of patients during infection and after recovery in Saudi Arabia

Dear valued reviewers,

On behalf of my fellow co-authors, we would like to acknowledge your efforts and time spent in revising our submitted paper. We strongly believe that such comments will strongly contribute to the scientific quality of the study. Please note that the changes based on your comments are highlighted in yellow.

Response to the suggested comments:

Reviewer 2

This study offers a modest added value to our understanding of patient experience and outcomes when coping with Covid-19. I prresent the following comments to help the authors bring the manuscritp to a point where it is ready for publication:

  1. Language editing - while generally clearly phrased the manuscript can ebenfit from having a professional english language editor "polishing" minor grammatical errors and cumbersome phrasing. 

Response: Please note that we plan to submit our final version of the manuscript to an English language editing company.

  1. I would reconsider the claim that COVID19 has become the most severe threat of our time. While socio-economic threats due to lockdowns were quite severe the pandemic was hardly a mass killer, with mortality rates lower than heart disease and cancer. 

 Response: Please not that it was dropped out. The introduction section was revised.

  1. In the introduction I would like to see a more thorough review of the concept of QOL and its importnance which are touched on very briefly in this current version. Then I'd liek to see a description of the EQ-5D, its rationale and structure, information about reliability and validity.(Concept of QOL, importnance, rationale and structure, reliability and validity)

Response: We have revised the whole introduction section to make it clearer. We have added these points:

  • The World Health Organization (WHO) referred to QoL as an individual's perception of their position in life, an extension of their environment in which they live, and in relation to their goals, expectations, standards and concerns.
  • HRQoL is a more specific measure that quantifies the physical and mental status of an individual or a group over time. This metric is generally used to represent the impact of an illness and its management on patients’ ability to live fulfilling lives and their overall life satisfaction.
  • HRQoL is an essential indicator of the burden of a disease that helps clinicians and policy-makers optimize patient care and public health decisions.
  • Many instruments have been developed to measure HRQoL including the five dimensional EuroQol (EQ-5D).
  • The EQ-5D is one of the most frequently used, preference-based measure of HRQoL worldwide that evaluates the HRQoL based on the patient’s perspective of their health.
  • EQ-5D is a valid and reliable measure that has been applied to countless disease areas and is the most commonly used tool in cost-utility analyses and for appraising healthcare interventions.
  • Since its development it has been validated across various settings and among different populations. However in terms of COVID-19 and in conservative communities such as in Saudi Arabia, using EQ-5D as a measure of HRQoL is scarce.
  1. In the last paragraph of the intro the authors touch numerous issues (natural sup0plements, nurtrition, other vaccines etc.) - but all of these factors REQUIRE a rationale - why are they important to include in your study model? Please add these factors as variables in your literature review and refer to why would they be interesting factors to include in your research model. 

Response: We really missed on this one. We have added the following: "In terms of infectious diseases such as COVID19, the experience might be different as other factors might play a role, such as previous vaccination (COVID-19, seasonal flu) and consumption of natural supplements such as drinking herbal drinks, honey or others which COVID-19 patients might perceive beneficial. In terms of consumption of natural supplements, sharing personal experiences can inform researchers, policymakers, as well as other individuals at risk of contracting COVID-19. Studies showed that herbal supplements and honey - which are commonly used in Saudi Arabia- have some antiviral activities. Please note that the consumption of natural supplements were not our primary factors to investigate. These were self-reported health interventions that participants perceived beneficial during the COVID-19 infection.

  1. The main methodological challenge here is that the authors try to assess the effect of CVID-19 infection on QOL without having a baseline measure. They use a 'post infection' measure as a substitute, but this is of course still a problem. At the very least the authors must discuss this issue and detail a rationale as to why using this alternative approach will be sufficient. 

Response: Yes we agree. The study design is actually an observational prospective study, in which a cohort of COVID-19 confirmed patients who visited the influenza clinic at King Abdulaziz Medical City (KAMC), Riyadh, Saudi Arabia, were enrolled between November and December of 2022. Baseline measures including the HRQoL were obtained at the clinic and then repeated after two weeks of recovery.

I would expect of course that most patioants with the diagnosis will report medium to low QOL - symptoms, hospitalization and the "bad reputation" of the disease alone are good reasons for that. I would absolutely expect a rise in QOL measure after gettijng better - but was QOL better before the disease? I really cannot tell. The increase in QOL may be an effect of the recovery. 

Response: You are absolutely right on this point. Our baseline measure of HRQoL was during sickness. Recruiting a healthy group, obtaining a baseline measure of HRQoL, then awaiting COVID-19 infection is impractical. Questioning COVID-19 patients about their HRQoL prior the disease is prone to recall bias, nevertheless desirability bias given as you said the "bad reputation" of the disease. The best option for us was to do same group analyses to test if the HRQoL enhanced after recovery or it remained the same. The change in the HRQoL might be different between subgroups and here we made our conclusions.

  1. Re-order the results: start with descriptive statistics for the sample: mease, sds, ranges, and reliability coefficients should be presented. Then - present the comparisons between during-post disease: simple differences and while controlling for background information (gender, age, education level,  medical hitory, for example). Then test your additional questions: the use of natural supplements, etc.  The current propostion tables are useless. Use ANOVA and ANCOVAto test for differences. Use multiple regression analysis to test for the eeffects of age, background diagnoses, gender and education on QOL during and after the disease. Drop the excessive tables and the graphs. Present the above mentioned analyses instead. 

Response: our data was revisited and we obtained statistical consultation. Descriptive statistics of categorical variables were presented in frequency and percentage, while continuous variables were presented in mean ± SD or median (inter-quartile range). Measures of internal consistency showed that the Cronbach's alpha of EQ-5D tool during COVID-19 and after recovery were 0.842 and 0.801 respectively. Same group analyses (within subgroups) using the Wilcoxon Signed Ranks Test was performed to compare the index scores of EQ-5D dimensions and EQ-VAS during the disease and at recovery given that the distribution of EQ-5D data is ordinal and commonly skewed (Arab-Zozani, 2020). Skewness coefficients ranged between 0.5 and 3.4.

The assumptions of Analyses of Covariance (ANCOVA) were violated due to the nature of skewed outcome and the presence of logic outliers. Therefore, between group analyses were performed using the Mann Whitney test by comparing the differences in the EQ-5D scores and EQ-VAS (recovery minus baseline). Higher differences in scores indicated a higher change in the HRQoL.

Variables statistically significant at the bivariate level of analyses (P<0.05) were accounted as potential predictors of changes in EQ-5D and EQ-VAS scores. Accordingly, two multiple linear regression models were constructed. To meet the regression assumptions, the skewness in EQ-5D scores was corrected by normalization (score-minimum value/range) to take a range zero-one and then log10 transformed (skewness coefficient became -0.150).  Homoscedasticity and  normality of the data were confirmed after examining the P-P plot and scatterplot. An absence of multi-collinearity was confirmed by examining the Variance Inflation Factor (VIF) values which were all under 5, indicating that the assumption is met. In terms of EQ-5D scores, no statistically significant interaction effects between various exposures were detected, except for being asthmatic and taking seasonal influenza vaccine. The model was fit, and the adjusted R2 was 33.5%. Insignificant variables such as curcuma/onion/peppermint/ wild thyme consumption, zinc deficiency, sleeping duration, diabetes, gender were dropped out which increased the adjusted R2 to 34.2%. In terms of EQ-VAS scores, no significant interaction effect was present. The model was fit, and the adjusted R2 was 33.5%. Insignificant variables such as onion/peppermint/ wild thyme consumption, smoking, sleeping duration, asthma, anemia, gender, age, and BCG vaccination were dropped out which increased the adjusted R2 to 38%.       

  1. Your discussion of your results is not fully based on yourt analyses. You used descriptive analyses and then draw conclusions as to associations and effects. This is unacceptable in scientific investigation. Use hypotheses testing as mentioned above to see if the effects you were looking for are indeed present and are statistically significant. 

Response: Please note that the discussion has been revised. It currently takes the following structure guided by two hypotheses: a. Recovery from COVID-19 will naturally enhance the HR-QoL among patients, b. Some factors might have contributed to a better improvement in the HR-QoL. In the first paragraph, we reported the findings of the first hypotheses after being tested "recovery from COVID-19 has enhanced the quality of life among Saudi patients as the scores of EQ-5D-5L index scores and EQ-VAS scores significantly increased after two weeks". This can be attributed to the absence of COVID-19 signs and symptoms. We then compared our findings to literature and provided some possible justifications. In the second and third paragraphs, we tested the second hypothesis to identify which factors led to a significant improvement in the HRQoL. In the fourth paragraph, we discussed the challenges in associating the consumption of natural supplements with the changes in HRQoL.

Round 2

Reviewer 2 Report

This revision is definitely an improvement upon the original version. I do , however, have a few additional issues I need the authors to check and revise as needed:

1. In the abstract the results do not specify which direction the associated factors work: for example I assume that normal body weight has a p[ositive association with a positive change in QOL, while being anaemic will have a negative association with the positive change - in other words - specify the "direction" of your effects. 

2. Sample description: sampling method is still missing -did you sample anyone arriving at this medical centre who met the criteria? during what period of time? how many men vs how many women? how many of them were previously healthy vs. previously infected? how many of them had history of chronic health conditions that could worsen their COVID19 symptoms? these are all missing from your description. 

3. I propose that the demographics appearing in the first chapter of the results will be moved to the sample description. 

4. The results chapter now makes a lot more sense to me. 

5. The discussion is more comprehensible now that also the rationale for supplements and health related behaviours has been spelled out. 

6. In the limitations - to me the greatest limitation is the lack of baseline measure before the exposure to COVID. Please discuss this challenge to the results validity. 

I believe that after addressing the above issues the manuscript will be ready for publication. 

Author Response

Response to Reviewers - ed2

(Journal: International Journal of Environmental Research and Public Health)

Study title:

Impact of COVID-19 on the Health-Related Quality of Life of patients during infection and after recovery in Saudi Arabia

Please note that the changes based on your comments are highlighted in yellow.

Response to the suggested comments:

Reviewer

Comments and Suggestions for Authors

This revision is definitely an improvement upon the original version. I do , however, have a few additional issues I need the authors to check and revise as needed:

  1. In the abstract the results do not specify which direction the associated factors work: for example I assume that normal body weight has a p[ositive association with a positive change in QOL, while being anaemic will have a negative association with the positive change - in other words - specify the "direction" of your effects.

Response:  Thank you for the advice. Please note that we have revised the results in the abstract.

" Multiple linear regression analyses showed that having a normal body weight, being employed, non anemic, and previously taking the BCG vaccine were positively associated with a higher change in the HRQoL. An interaction between being asthmatic and taking the influenza vaccine positively predicted a lower change in the HRQoL. Having a normal weight positively predicted a higher change in the perceived health state after recovery. Increasing the consumption of natural supplements (honey and curcuma) did not improve the HRQoL and the perceived health state."

  1. Sample description: sampling method is still missing -did you sample anyone arriving at this medical centre who met the criteria? during what period of time? how many men vs how many women? how many of them were previously healthy vs. previously infected? how many of them had history of chronic health conditions that could worsen their COVID19 symptoms? these are all missing from your description.

Response: Please note that we have added these details to the methods section.

  • sampling method is still missing -did you sample anyone arriving at this medical centre who met the criteria?

By convenience, COVID-19 positive cases circumstantially present at the targeted setting during the study period were screened for eligibility then invited to participate.

  • during what period of time?

November and December of 2022

  • how many men vs how many women?

Two thirds of the sample were women (n=259).

  • how many of them were previously healthy vs. previously infected?

When asked about the number of times they had an episode of cold symptoms in the past six months, 352 participants (90.5%) reported having one to three episodes, yet non were confirmed as COVID-19 through a PCR test.

  • how many of them had history of chronic health conditions that could worsen their COVID19 symptoms?

Almost half of the participants (55.8%) had at least one chronic disease.

  1. I propose that the demographics appearing in the first chapter of the results will be moved to the sample description.

Response: We agree since it is a prospective study. We have reallocated this part to the methods sections.

  1. The results chapter now makes a lot more sense to me.

Response: Thank you

  1. The discussion is more comprehensible now that also the rationale for supplements and health related behaviours has been spelled out.

Response: Thank you

  1. In the limitations - to me the greatest limitation is the lack of baseline measure before the exposure to COVID. Please discuss this challenge to the results validity.

Response: We agree. We have added teh following to the limitations section:

" Finally, one of the challenges in this study was the lack of a baseline measure before the exposure to COVID-19 infection. Unfortunately, recruiting a healthy group, obtaining a baseline measure of HRQoL, then awaiting a COVID-19 infection is a lengthy process and prone to a loss in follow-up. Questioning COVID-19 patients about their HRQoL prior the COVID-19 infection is also prone to recall bias. Despite these limitations, factors significantly associated with changes in the HRQoL and perceived health status in this study remain valid within the context of COVID-19 infections and Saudi Arabian population. "

Thank you
